# Sarcopenia in the Oldest-Old Adults in the Capital of Brazil: Prevalence and Its Associated Risk Factors

**DOI:** 10.3390/nu16233976

**Published:** 2024-11-21

**Authors:** Hudson Azevedo Pinheiro, Ruth Losada de Menezes, Camila Kellen de Souza Cardoso, Rômulo Roosevelt da Silva Filho, Farah Registre, César de Oliveira, Erika Aparecida Silveira

**Affiliations:** 1Health Department of the Federal District, Brasília 72010-120, Brazil; hudsonap@gmail.com; 2Study, Research, and Extension Group on Functional Health and Aging, Interdisciplinary Center on Aging, Postgraduate Program in Health Sciences, Institute of Tropical Pathology and Public Health, Federal University of Goiás, Goiania 74605-050, Brazil; ruthlosada@ufg.br; 3Postgraduate Health Science Program, Medicina Faculty, Federal University of Goiás, Goiania 74690-900, Brazil; camilacardoso_nut@hotmail.com (C.K.d.S.C.); romulordsf@gmail.com (R.R.d.S.F.); farah.registre@yahoo.fr (F.R.); 4Department of Epidemiology & Public Health, Institute of Epidemiology & Health Care, University College London, London WC1E 6BT, UK; c.oliveira@ucl.ac.uk

**Keywords:** sarcopenia, thinness, risk factors, activities of daily living, prevalence, aged 80 and over

## Abstract

Background: In light of the demographic context in which the older adult population is prominent, sarcopenia emerges as a significant concern for the health of these individuals. Aim: To assess the frequency of sarcopenia and severe sarcopenia and the associated risk factors in the oldest adults living in the community. Methods: There were 399 participants aged 80 or older, of both sexes, using primary health care services in the metropolitan area of Brasília, Brazil. Sarcopenia was evaluated based on European Working Group on Sarcopenia in Older People 2 (EWGSOP2). Muscle mass was measured by calf circumference, muscle function by handgrip strength, and muscle performance by gait speed. Clinical and socioeconomic variables, comorbidities, falls, and urinary incontinence were collected. The prevalence of sarcopenia was calculated with a 95% (IC) prevalence. Multivariate Poisson regression analysis was performed in Stata, with *p* < 5%. Results: Among participants, 78.2% were women. Prevalence of pre-sarcopenia was 31.8%, sarcopenia 3.3%, and severe sarcopenia 25.1%. After multivariate regression, sarcopenia was associated with the female sex, low weight, and a dependency for activities of daily living (ADLs). Similarly, severe sarcopenia remained associated with female sex, low weight, and an ADLs dependency. Sarcopenia and severe sarcopenia were not associated with the level of education, marital status, income, physical activity, medications, falls, nor comorbidities. Conclusions: A quarter of older adults had severe sarcopenia. Sarcopenia and severe sarcopenia were associated with being a woman, being low weight, and have an ADLs dependence.

## 1. Introduction

As life expectancy continues to rise globally, the proportion of older adults is also increasing. This demographic trend is observed in many countries including Brazil, where the population aged 80 and over is growing fast [1,2].

According to the United Nations Organization (ONU) (2022), Brazil’s population aged 65 or older may increase by approximately 20% over the next five decades [3]. The demographic shift towards an older population presents significant challenges for health systems, social care services, and the scientific community. This underscores the critical need to understand and address the economic, health, social, and psychological needs in later life [4,5,6].

In light of the demographic context in which the older adult population is prominent, sarcopenia emerges as a significant concern for the health of these individuals. With increasing age, the decline in muscle mass and function becomes inevitable, gradual, and continuous. This decline results in limitations in daily activities, an increased risk of falls and fractures, and a poorer prognosis for several chronic diseases [7,8,9].

There are still few studies of community-dwelling that have included people aged 80 years or older which investigate severe sarcopenia, sarcopenia and pre-sarcopenia and its associated factors, especially in Latin American countries, including Brazil. Sarcopenia is a geriatric syndrome, the impact of which on disabilities and mortality, however, needs to be investigated further in older adults at or exceeding 80 years old [10,11,12]. The few Brazilian studies mostly investigate this population in Sao Paulo [13].

Our study is the first with this specific population (80+) to be conducted in Brasilia, the capital of Brazil, the third largest metropolitan region. To investigate these outcomes in oldest or older adults brings relevant clinical and public health information. In Brazil, a study found that 16.1% of women and 14.4% of men aged 60 and over in São Paulo were affected by sarcopenia; the most significant risk factors for sarcopenia are age, cognitive decline, and the risk of malnutrition [13]. The present study chose not to explore cognitive declines because it understands that neurological problems and dementia syndromes in themselves are determining factors for functional dependence and to demystify that not every person over 80 years of age has cognitive deficits or even dementia senile.

In Fortaleza, Ceará, Brazil, the prevalence of probable sarcopenia was 25.52%, with 11.98% cases of sarcopenia and 9.90% of severe sarcopenia in individuals aged 60 years or older. The results indicated that probable sarcopenia was more prevalent in males and in patients receiving multiple medications, while calf circumference below 31 cm was more frequent in patients with sarcopenia and severe sarcopenia. Furthermore, the presence of osteoporosis was more common in cases of severe sarcopenia [14].

Despite numerous studies on sarcopenia, no specific research has been identified that addresses the prevalence and risk factors of sarcopenia and severe sarcopenia in older adults over the age of 80 in Brazil; with the progressive aging of the population, especially in more vulnerable regions (socially, economically, and environmentally), the identification of the sarcopenia phenotype becomes necessary as a public policy within the Unified Health System (UHS), the Brazilian program that finances government actions relating to health policies aiming to minimize impacts such as exacerbation of chronic diseases, dependence on carrying out activities of daily life (ADLs), caregiver burden, and high costs for services [15,16].

This paper aims to contribute to a more comprehensive understanding of this condition in Brazil and improve the quality of life in later life. Additionally, the findings will inform the development of more effective public health policies and clinical practices.

## 2. Materials and Methods

### 2.1. Study Design and Populations

This is a cross-sectional study carried out between September 2015 and December 2018 with community-dwelling older adults assisted by the UHS in the southwest region of Brasilia (the third largest Brazilian metropolis), from eight basic health units (BHU) and a family clinic, Federal District, Brazil, with an estimated population of 828,703 inhabitants. We know that individuals from 80 years and older in a country with a high rate of violence do not accept unfamiliar strangers into their homes. To minimize the recuses in our study and reduce bias, we include individuals who use the BHU. A Brazilian study reveals that 46.2% of the population over 60 years in age uses the health services [17]. Approximately 20% of that population is 80 or older, and their per capita income is around USD 315.04 [18].

The inclusion criterion adopted was older adults referred by BHU to the reference clinic in geriatrics and gerontology, located in Taguatinga, Brasilia/Federal District, Brazil. These patients were in conditions of vulnerability diagnosed by a doctor or a family health strategy team according to the criteria established by The Elderly Person’s Health Booklet in the context of Primary Care [19] or in consultation by an interdisciplinary team composed of a nurse, a physiotherapist, and a nutritionist who was duly trained for this purpose and who recorded the medications in use, the socioeconomic characteristics, and the clinical comorbidities previously diagnosed. They also recorded in the electronic medical record, in addition to specifically related complaints of vulnerability such as the number of falls in the previous year, if any regular physical activity was performed (i.e., at least 150 min of weekly practice of any modality), urinary incontinence, dependence in ADLs, and sarcopenia status.

The exclusion criteria for this research were those older adults who presented sequelae of neurological diseases (cerebrovascular disease, Parkinson’s disease, among others) or a cognitive deficit assessed by the mini-mental state (MMS) [20], in addition to amputees, since such injuries are related to the increased risk of dependence for ADLs and the increased risk of immobility.

During the interdisciplinary consultation, the following steps were completed sequentially, with the participation of family members and/or caregivers of these elderly individuals. Initially, an interview was conducted to survey a profile with issues related to income, marital status, and education, in addition to the ratification of information contained in the reason for referral, such as the presence of urinary incontinence, one or more falls in the last six months, or number of medications in use.

The study was approved by the Ethics and Research Committee of the Health Education and Research Foundation under opinion number 1,128,355/2015. All participants and their family members or caregivers have provided a signed and informed consent form.

### 2.2. Measurements

#### 2.2.1. Anthropometric Variables

Anthropometric variables were measured by height (cm) and weight (kg) using a scale with a stadiometer brand Filizola^®^ (São Paulo, Brazil) to subsequently calculate the body mass index (BMI), using the Lipschitzl recommendation; this classified the older adults with values less than 22 kg/cm^2^ as low weight, the older adults with values between 22 and 27 kg/cm^2^ as eutrophic subjects, and the older adults with a BMI greater than 27 kg/cm^2^ as excess weight, these measures being more sensitive for public health [21].

#### 2.2.2. Sarcopenia

To classify whether the older adults were sarcopenic, the criteria recommended by the EWGSOP2 were used, where subjects who presented reduced muscle strength and reduced quality of muscle mass were classified as sarcopenic. To be defined as severe sarcopenic, older adults should also present reduced physical performance, and pre-sarcopenia adults present only reduced muscle mass or only reduced muscle quality [22].

To measure muscle strength, the handgrip strength test was measured with a hydraulic dynamometer JAMAR^®^ (São Paulo, Brazil). Three measurements with an interval were performed in the dominant hand. One minute between them is considered sarcopenic if they present values lower than 27 kg/F for men and 16 kg/F for women [23].

The quality of muscle mass was measured by the circumference of the calf with the participant sitting on a chair, with legs relaxed, feet flat on the floor, and knees bent at 90°. After identifying the most protruding region of the legs through an inelastic measuring tape, the perimetry was gauged, and individuals with values equal to or less than 33 cm for women and 34 cm for men were considered at risk for sarcopenia [24,25].

Finally, the physical performance was measured by the usual gait speed test performed in a corridor, where participants were instructed to walk at their usual speed, being able to use an auxiliary device for locomotion. The time taken to move three meters was measured, after the acceleration and deceleration time were disregarded; a speed slower than 0.8 m/s was considered a risk. All functional capacity tests were conducted by a qualified examiner [26].

#### 2.2.3. The Activity of Daily Living Assessment

The Barthel index was applied to assess the degree of functional independence, using the version validated and cross-culturally adapted for the Brazilian population, whose cutoff point greater than or equal to 60 suggests the subjects are independent for ADLs [27].

### 2.3. Statistical Analyses

The statistical analyses were performed in the software Stata 12.0. The outcomes of this study were sarcopenia and severe sarcopenia. We estimated the prevalence ratio with their 95% confidence intervals and associated risk factors according to all sociodemographic and clinical variables.

All variables with *p*-value ≤ 0.20 in bivariate Poisson regression were included in the multivariable Poisson regression to control for potential confounders. The criteria to maintain variables on the final regression model as *p*-value ≤ 0.05 denominated as the adjusted model on the tables.

## 3. Results

Our analytical sample included 399 older adults aged 80 to 104 years. The mean age was 87.34 years (SD = 5.22), the mean BMI was 25.71 kg/m^2^ (SD = 5.32), the mean calf circumference was 31.77 cm (SD = 4.11), and the mean hand grip strength was 17.73 kg/F (SD = 6.23).

The prevalence of pre-sarcopenia was 31.8% (95% CI = 26.9–36.1), sarcopenia was 3.3% (CI 95%= 1.5–5.1), and severe sarcopenia 25.1% (95% CI = 20.8–29.6), represented in Figure 1. The prevalence of pre-sarcopenia and severe sarcopenia were statistically different between the sexes. The prevalence of pre-sarcopenia was significantly higher in men than in women (*p*-value = 0.013). Severe sarcopenia was more prevalent in women (28.5%) than in men (12.6%) (*p*-value = 0.03).

The prevalence of non-sarcopenia, pre-sarcopenia, sarcopenia, and severe sarcopenia according to sociodemographic characteristics and physical activity were statistically different by sex, marital status, monthly income, and activities of daily living (ADLs) dependence. The data are available in Table 1.

The prevalence of pre-sarcopenia, sarcopenia, and severe sarcopenia and their association with health variables in community-dwelling older adults were statistically different in individuals with low weight, ADLs independence, and COPD. The data are available in Table 2.

Sex, marital status, low weight, ADLs dependency, and depression were statistically associated with sarcopenia in the unadjusted regression analysis. However, multivariate regression analyses included additional factors such as level of education, COPD, urinary incontinence, and falls. After the multivariate analysis, sarcopenia showed statistically significant associations with the female sex (PR 2.34, 95% CI 1.35–4.07), a low weight status (PR 1.93, 95% CI 1.45–2.59), and an ADLs dependence (PR 1.94, 95% CI 1.35–2.79). The data are available in Table 3.

Concerning severe sarcopenia, the associated variables were the female sex, marital status, underweight, ADLs dependency, and depression in the simple regression analysis. Level of education, falls, COPD and urinary incontinence have also been included in the multivariate Poisson regression. After the multivariable analysis, the associated variables with severe sarcopenia were female sex PR 2.32 (IC 95% 1.29–4.14), underweight PR 2.00 (IC 95% 1.46–2.75), and ADLs dependency PR 2.13 (IC 95% 1.47–3.10). The data are available in Table 4.

## 4. Discussion

This study not only represents the first comprehensive investigation into the prevalence of sarcopenia and severe sarcopenia in older Brazilians aged 80 and older but also sheds light on the associated risk factors, paving the way for targeted interventions and improved public health strategies for this population.

The results showed a prevalence of possible sarcopenia, sarcopenia, and severe sarcopenia in older adults aged 80 or older at rates of 31.8%, 3.3%, and 25.1%, respectively. When compared with international studies, the rates vary.

In China, 38.5% of older adults over 80 years of age had possible sarcopenia, 18.6% had sarcopenia, and 8.0% had severe sarcopenia [10]. In Chile, the prevalence among people aged 80 years and older was 38.5% [28]. In contrast, in Finland, older adult men had lower rates: 4.8% probable sarcopenia and 2.7% confirmed sarcopenia [14].

The differences in prevalence rates may be due to population characteristics, highlighting the importance of addressing this issue in healthcare and promoting strategies to maintain muscle health as older adults.

In Brazilian studies, the prevalence of sarcopenia in primary care showed that males were more likely to develop pre-sarcopenia, and the female sex was a risk factor for sarcopenia, since, due to hormonal changes beginning from the age of 50, there is an accelerated loss of strength in women; there is also increased risk for individuals aged over 76 years [29,30].

The study found a higher prevalence of pre-sarcopenia in men and severe sarcopenia in women (28.5% vs. 12.6%). This is consistent with the findings from Sousa et al., [29] showing that probable sarcopenia is more prevalent in men. In contrast, Wu et al. [10] reported higher possible sarcopenia in women (40.7% vs. 36.3%), while Lera et al. [28] found equal sarcopenia prevalence (19.1%) in both sexes in Chile. These variations could be attributed to age, lifestyle, and genetic factors.

Our study has assessed the prevalence of different stages of sarcopenia, analysing sociodemographic data and physical activity levels. Significant differences were observed concerning gender, marital status, monthly income, and dependence to perform ADLs.

A systematic review focusing on community-dwelling older adults aged 60 and over identified the following factors associated with sarcopenia: advanced age, marital status, ADLs, and low weight. Nevertheless, no significant associations were found between sarcopenia and either male or female sex in this study [6]. Concerning monthly income, no significant differences were observed between groups of older adults with varying degrees of sarcopenia [13].

Our findings revealed statistically significant differences in the prevalence of pre-sarcopenia, sarcopenia, and severe sarcopenia, as well as their relationships with health variables, especially related to low weight, independence for ADLs, and COPD. Santos et al. [31] noted a higher prevalence of sarcopenia in underweight older adults aged 80 to 84 years. On the other hand, Jones et al. [32] linked increased sarcopenia prevalence with COPD progression in older adults with an average age of 70.4 years.

According to Wu X et al., [10] having a history of chronic lung diseases was associated with a higher risk of possible sarcopenia. However, no identified association was found between the severity of COPD and the prevalence of sarcopenia in the study by Jones et al. [32]. These results highlight the importance of considering various health variables when evaluating sarcopenia in older adults.

The results of the multivariate analysis indicated that female sex, low weight, and ADLs remained statistically significant after controlling for other variables. These factors demonstrated associations with sarcopenia and severe sarcopenia, indicating their importance even after consideration of additional factors such as education, COPD, urinary incontinence, and falls.

Previous studies, such as Yen et al., [33] also identified female gender as a risk factor for severe sarcopenia. Additionally, Oliveira et al. [34] found that female sex is associated with sarcopenia in institutionalized older individuals. These findings underscore the significance of gender in the risk profile for sarcopenia and suggest that targeted interventions for women may be necessary.

Our findings align with Sri-On et al. [35] who showed that low weight is linked to sarcopenia and severe sarcopenia in individuals aged 70 and above. Another study indicated that malnutrition is associated with an approximately four times greater risk of developing sarcopenia or severe sarcopenia with advancing age due to a poorly balanced diet and a reduction in micro- and macronutrients [36].

Conversely, in individuals with severe obesity, the complexity of identifying negative variables for muscle health is compounded by factors such as chronic inflammation, insulin resistance, dysphagia, and low physical activity [37,38]. These elements are crucial for muscle mass loss and reduced strength, which can give rise to further complications.

Regarding sedentary lifestyle, Brazilian studies found that those elderly people who had higher income regularly practiced some physical activity while low-income subjects considered occupational activity as physical activity. In addition, subjects with low income tend to adopt other less healthy habits, such as smoking and alcohol consumption, due to a lack of information, contributing to the development of comorbidities or chronic diseases regardless of sex [13,39].

Concerning ADLs, a survey of older adults indicated that sarcopenia was directly linked to greater disability in ADLs and lower physical functionality. Older individuals with sarcopenia were twice as likely to face limitations in ADLs compared to those without this condition. Furthermore, sarcopenia was associated with weakness in the lower limbs, which caused difficulties such as bending, kneeling, lifting loads above 5 kg, and walking 400 m [8,40,41].

The association between sarcopenia and dependence on ADLs results in increased care costs, greater caregiver burden, and a greater risk of hospitalization [42,43,44,45]. Therefore, it is essential to identify sarcopenia early and implement effective interventions to enhance muscle strength and function in older adults, mainly in primary care which is responsible for ensuring access to basic healthcare and preventing diseases such as sarcopenia through physical, nutritional, and clinical health education actions throughout the course of life and close to one’s home [46].

This study has limitations that should be acknowledged. The lack of data on lifestyle such as smoking status, alcohol consumption, leisure activities, ethnicity, religion, and biochemical markers could be a potential limitation. These factors may potentially impact the development of sarcopenia and should be considered in future research involving older individuals. By including these variables, we can better understand their impact on the sarcopenia process.

Additionally, our study excluded individuals with pre-existing physical or mental limitations, assuming they already had reduced mobility and increased dependence on ADLs. However, investigating sarcopenia in these populations can provide valuable insights into how such conditions influence sedentary behaviours and lifestyles.

It is essential to note that the prevalence and associated factors of sarcopenia may change over the lifespan. Recent evidence indicates that the risk factors commonly associated with sarcopenia in later life may not apply to the oldest elderly individuals living in the community. For instance, the correlation between physical activity levels and daily functioning with sarcopenia may not be as straightforward as previously thought. Indeed, being overweight may even offer protection against this condition, contrary to trends observed in older adults more broadly. Nevertheless, further research is required to investigate the risk factors for sarcopenia and severe sarcopenia, particularly in long-lived elderly populations, to inform the development of more effective preventive strategies and interventions.

## 5. Conclusions

The high prevalence of severe sarcopenia affects approximately a quarter of the oldest seniors. Several factors have been associated with sarcopenia and severe sarcopenia risk, including being female, low weight, and ADLs dependency. This research highlights the pressing requirement for public health policies and strategies to reduce the effects of sarcopenia and encourage a healthy and active ageing process.

## Figures and Tables

**Figure 1 nutrients-16-03976-f001:**
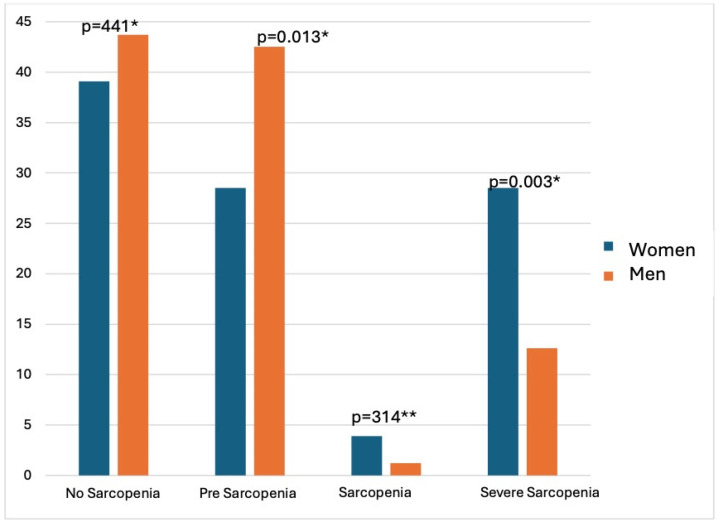
Prevalence of pre-sarcopenia, sarcopenia, and severe sarcopenia by sex in community-dwelling older adults (*n* = 399). Notes: * Chi-square test; ** Fisher’s exact test.

**Table 1 nutrients-16-03976-t001:** Prevalence of pre-sarcopenia, sarcopenia, and severe sarcopenia and their associations with sociodemographic variables and physical activity in community-dwelling older adults (*n* = 399).

Variables	n (%)	No Sarcopenia (%)	Pre-Sarcopenia (%)	Sarcopenia (%)	Severe Sarcopenia (%)	*p*
**Sex**						
Female	312 (78.2)	122 (39.1)	89 (28.5)	12 (3.8)	89 (28.5)	0.004 *
Male	87 (21.8)	38 (43.7)	37 (42.2)	1 (1.2)	11 (12.6)	
**Level of education**						
Illiterate	106 (26.6)	38 (35.8)	31 (29.3)	1 (0.9)	36 (33.9)	0.113 *
≤6 years	212 (53.1)	86 (40.6)	70 (33.0)	11 (5.2)	45 (21.2)	
>7 years	81 (20.3))	36 (44.4)	25 (30.9)	1 (1.2)	19 (23.5)	
**Marital status**						
Not married/divorced	50 (12.6)	22 (44.0)	15 (30)	0 (0.0)	13 (26.0)	0.022 *
Married	111 (27.9)	49 (44.1)	42 (37.8)	5 (4.5)	15 (13.5)	
Widower	237 (59.5)	89 (37.5)	68 (28.7)	8 (3.4)	72 (30.4)	
**Monthly income per person**						
≤U$200	106 (26.6)	48 (45.3)	30 (28.3)	0 (0.0)	28 (26.4)	0.036 *
U$200–400	227 (57.0)	81 (35.7)	82 (36.1)	11 (4.9)	53 (23.3)	
≥400	65 (16.3)	30 (46.2)	14 (21.5)	2 (3.1)	19 (29.2)	
**Physical Activity Status (>150 min/week)**						
No	216 (54.1)	80 (37.0)	75 (34.7)	6 (2.8)	55 (25.5)	0.401 **
Yes	183 (45.9)	80 (43.7)	51 (27.8)	7 (3.8)	45 (24.6)	

Note: * Fisher’s exact test; ** Chi-square test.

**Table 2 nutrients-16-03976-t002:** Prevalence of pre-sarcopenia, sarcopenia, and severe sarcopenia and their associations with health variables in community-dwelling older adults (*n* = 399).

Variables	*n* (%)	No Sarcopenia (%)	Pre-Sarcopenia (%)	Sarcopenia (%)	Severe Sarcopenia (%)	*p*
**Low weight**						
No	297 (74.4)	153 (51.5)	78 (26.7)	8 (2.6)	58 (19.5)	*p* < 0.001 *
Yes	102 (25.6)	7 (6.9)	48 (47.1)	5 (4.9)	42 (41.2)	
**Falls**						
No	224 (56.3)	92 (41.1)	75 (33.5)	8 (3.6)	49 (21.9)	0.377 *
Yes	174 (43.7)	68 (39.1)	50 (28.7)	5 (2.9)	51 (29.3)	
**Femur fractures**						
No	386 (96.7)	157 (40.7)	121 (31.3)	13 (3.4)	95 (24.6)	0.465 *
Yes	13 (3.3)	3 (23.1)	5 (38.5)	0 (0.0)	5 (38.5)	
**Independence for ADLs**						
No	25 (6.3)	7 (28.0)	4 (16.0)	0 (0.0)	14 (56.0)	0.008 *
Yes	374 (93.7)	153 (40.9)	122 (32.6)	13 (3.5)	86 (23.1)	
**Arterial hypertension**						
No	62 (15.6)	18 (29.0)	26 (41.9)	3 (4.8)	15 (24.2)	0.127 *
Yes	336 (84.4)	141 (41.9)	100 (29.8)	10 (3.0)	85 (25.3)	
**Cardiopathy**						
No	297 (74.4)	124 (41.7)	92 (31.0)	11 (3.7)	70 (23.6)	0.478 *
Yes	102 (25.6)	36 (35.3)	34 (33.3)	2 (2.0)	30 (29.4)	
**Diabetes**						
No	253 (63.4)	102 (40.3)	81 (32.0)	8 (3.2)	62 (24.5)	0.980 *
Yes	146 (36.6)	58 (39.7)	45 (30.8)	5 (3.4)	38 (26.0)	
**Depression**						
No	220 (55.3)	89 (40.5)	72 (32.7)	5 (2.3)	54 (24.5)	0.732 *
Yes	178 (44.7)	70 (39.3)	54 (30.3)	8 (4.5)	46 (25.8)	
**Arthrosis**						
No	210 (52.6)	77 (36.7)	73 (34.8)	6 (2.9)	55 (25.7)	0.389 **
Yes	189 (47.4)	83 (43.9)	53 (28.0)	7 (3.7)	46 (24.3)	
**Hypothyroidism**						
No	341 (85.5)	137 (40.2)	107 (32.4)	10 (2.9)	87 (25.5)	0.739 *
Yes	58 (14.5)	23 (39.7)	19 (32.8)	3 (5.2)	13 (22.4)	
**Chronic obstructive pulmonary disease (COPD)**						
No	391 (98.0)	160 (40.9)	122 (31.2)	13 (3.3)	96 (24.5)	0.048 *
Yes	8 (2.0)	0 (0.0)	4 (50.0)	0 (0.0)	4 (50.0)	
**Urinary incontinence**						
No	166 (41.6)	66 (39.8)	59 (35.5)	6 (3.6)	35 (21.2)	0.344 **
Yes	233 (58.4)	94 (40.3)	67 (28.8)	7 (3.0)	65 (27.9)	
**Medications (number)**						
† 0–4	158 (39.6)	60 (37.9)	52 (32.9)	7 (4.4)	39 (24.7)	0.668 **
5 or more	241 (60.4)	100 (41.5)	74 (30.7)	6 (2.5)	61 (25.3)	

Note: * Fisher’s exact test. ** Chi-square test; † 10 older adults did not use medication.

**Table 3 nutrients-16-03976-t003:** Association of sarcopenia in community-dwelling older adults, simple and multivariate Poison regression (*n* = 399).

Variables	Simple Regression		Multivariate Regression	
	RP (CI 95%)	*p*-Value (Wald)	RP (CI 95%)	*p*-Value
**Sex**		0.0024		
Female	2.35 (1.35–4.07)		2.34	0.002
Male	1		1	
**Level of education**		0.1930	-	
Illiterate	1.41 (0.89–2.24)		-	-
≤6 years	1.07 (0.69–1.66)		-	-
>7 years	1			
**Marital status**		0.0153		
Not married/divorced	1.44 (0.78–2.67)		-	-
Married	1		-	-
Widower	1.87 (1.21–2.89)			
**Monthly income per person**		0.6991		
≤U$200	1		-	-
U$200–400	1.07 (0.73–1.56)		-	-
≥400	1.22 (0.76–1.97)		-	-
**Physical Activity status**		0.9693	-	-
No	1		-	-
Yes	1.00 (0.73–1.38)			
**Medications (number)**		0.7758	-	-
0–4	1		-	-
5 or more	0.95 (0.69–1.31)			-
**Low weight**		0.0000		
No	1		1	
Yes	2.07 (1.54–2.79)		1.93 (1.45–2.59)	0.000
**Falls**		0.1394	-	
No	1			
Yes	1.26 (0.93–1.73)			
**Femur fractures**		0.3776		
No	1			
Yes	1.37 (0.68–2.79)			
**Independence for ADLs**		0.0001	-	0.000
No	2.1 (1.44–3.11)		1.94 (1.35–2.79)	
Yes	1		1	
**Arterial hypertension**		0.9029		
No	1.03 (0.67–1.57)		-	-
Yes	1		-	-
**Cardiopathy**		0.4226		
No	1		-	-
Yes	1.15 (0.82–1.62)		-	-
**Diabetes**		0.7027		
No	1		-	-
Yes	1.06 (0.77–1.47)		-	-
**Depression**		0.0000		
No	1		-	-
Yes	1.13 (0.83–1.54)		-	-
**Arthrosis**		0.9069		
No	1.02 (0.74–1.39)		-	-
Yes	1		-	-
**Hypothyroidism**		0.8937		
No	1.03 (0.66–1.62)		-	-
Yes	1		-	-
**Chronic obstructive pulmonary disease (COPD)**		0.1078		
No	1		-	-
Yes	1.79 (0.88–3.65)		-	-
**Urinary incontinence**		0.1809		
No	1		-	-
Yes	1.25 (0.90–1.74)		-	-

**Table 4 nutrients-16-03976-t004:** Association of severe sarcopenia in community-dwelling older adults, simple and multivariate Poison regression (*n* = 399).

Variables	Simple Regression		Multivariate Regression	
	RP (CI 95%)	*p*-Value (Wald)	RP (CI 95%)	*p*-Value
**Sex**		0.0044		0.004
Female	2.32 (1.30–4.14)		2.32 (1.29–4.14)	
Male	1		1	
**Level of education**		0.0624		
Illiterate	1.53 (1.06–2.22)		-	-
≤6 years	1		-	-
>7 years	1.06 (0.66–1.69)		-	-
**Marital status**		0.0082		
Not married/divorced	1.84 (0.95–3.56)		-	-
Married	1		-	-
Widower	2.22 (1.34–3.69)			
**Monthly income per person**		0.6572		
≤U$200	1.08 (0.72–1.60)		-	-
U$200–400	1		-	-
≥400	1.23 (0.79–1.91)		-	-
**Physical Activity status**		0.890		
No	0.97 (0.69–1.37)		-	-
Yes	1		-	-
**Medications (number)**		0.9774		
0–4	1		-	-
5 or more	1.00 (0.71–1.42)		-	-
**Low weight**		0.0000		
No	1		1	
Yes	2.16 (1.56–2.98)		2.00 (1.46–2.75)	0.000
**Falls**		0.0969		
No	1		-	-
Yes	1.33 (0.95–1.86)		-	-
**Femur fractures**		0.2552		
No	1			
Yes	1.51 (0.74–3.07)			
**Independence for ADLs**		0.0000		
No	2.35 (1.58–3.48)		2.13 (1.47–3.10)	0.000
Yes	1		1	
**Arterial hypertension**		0.9169	-	-
No	1		-	-
Yes	0.97 (0.61–1.57)			
**Cardiopathy**		0.2712		
No	1		-	-
Yes	1.22 (0.85–1.76)		-	-
**Diabetes**		0.7222		
No	1		-	-
Yes	1.06 (0.75–1.51)		-	-
**Depression**		0.0000		
No	1		-	-
Yes	1.08 (1.02–1.51)		-	-
**Arthrosis**		0.789		
No	1.05 (0.75–1.47)		-	-
Yes	1		-	-
**Hypothyroidism**		0.683		
No	1.11 (0.67–1.85)		-	-
Yes	1		-	-
**Chronic obstructive pulmonary disease (COPD)**		0.063		
No	0		-	-
Yes	1.97 (0.96–4.02)		-	-
**Urinary incontinence**		0.134		
No	1		-	-
Yes	1.31 (0.92–1.88)		-	-

## Data Availability

The Excel file with the database can be made available by the authors upon request.

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
