# Peer review of "Sarcopenia in the Oldest-Old Adults in the Capital of Brazil: Prevalence and Its Associated Risk Factors"

_nutrients, 2024, doi:10.3390/nu16233976_

Round 1
Reviewer 1 Report
Comments and Suggestions for Authors
The authors investigated the characteristics of community-dwelling people aged 80 years or older who suffer from sarcopenia. The purpose of this study is understandable, but many similar studies have been reported in the past. The background to this study, the significance of this study, and its characteristics (novelty) that are different from past studies should be fully explained to the reader. In addition, there are some points that need to be revised, so please see below.
Line 40, 230, 261
Citations should be displayed in [ ].
Line 67-69
Why was it necessary to target elderly people aged 80 years or older? What clinical benefits are there to investigating people aged 80 years or older?
Line 72
There are two periods.
Line 116-135
Please add an explanation of the diagnostic criteria for presarcopenia, sarcopenia, and severe sarcopenia.
Line 141-149
Two-group and three-group comparisons were performed when selecting risk factors, but they are not described here. This information should be included.
Results
The insertion position of figures and tables must be indicated in the text.
Line 240-245
Please provide the references.
Other
A similar study conducted in Brazil (reference 12) has found that cognitive function and malnutrition are risk factors. However, these were not examined in this study.
Author Response
We thank the reviewers for their thoughtful and in-depth comments concerning our manuscript. Your suggestions helped us improve the quality of our paper. We carefully considered every comment and made the appropriate changes, which are highlighted in red font.
Our point-by-point responses are also noted below.
The authors investigated the characteristics of community-dwelling people aged 80 years or older who suffer from sarcopenia. The purpose of this study is understandable, but many similar studies have been reported in the past. The background to this study, the significance of this study, and its characteristics (novelty) that are different from past studies should be fully explained to the reader. In addition, there are some points that need to be revised, so please see below.
There are still few studies (10, 11, 12) of community-dwelling that had included people aged 80 years or older which had investigated severe sarcopenia, sarcopenia and pre-sarcopenia and its associated factors, mainly in Latin American countries, including Brazil. Sarcopenia is a geriatric syndrome that impact in disabilities and mortality (reference) however it needs to be more investigated in older adults with 80 years and more. The few Brazilian studies mostly investigate this population in Sao Paulo (13). Our study is the first with this specific population (80+) conducted in Brasilia, the capital of Brazil, the third largest metropolitan region. To investigate these outcomes in oldest-older adults brings relevant clinical and public health information.
We also included this paragraph on introduction lines 50-64.
Line 40, 230, 261 Citations should be displayed in [ ]. We had corrected and highlighted in the text.
Line 67-69: Why was it necessary to target elderly people aged 80 years or older? What clinical benefits are there to investigating people aged 80 years or older?
It is so relevant to investigate individuals 80 years and older due the progressive aging of the world population and the few studies exploring this specific population, especially in Latin America and in vulnerable regions (social, economic and environmental). The knowledge of the occurrence of sarcopenia and associated factors will help to understand the impact of this outcome in 80 years and older individuals and then to elaborate adequate clinical protocols to prevent it and the worsen clinical consequences such us fail and frail syndrome. The identification of the sarcopenia phenotype also is necessary for policy makers to development actions to minimize impacts such as exacerbation of chronic diseases, dependence on carrying out activities of daily life (ADLs), caregiver burden and high costs for services
Line 72 There are two periods.
We corrected it.
Line 116-135 Please add an explanation of the diagnostic criteria for presarcopenia, sarcopenia, and severe sarcopenia.
We had improved it on this version of our manuscript and you can see in lines 123 and 142, the criteria for diagnosing pre-sarcopenia, sarcopenia and severe sarcopenia. Their respective diagnostic criteria’s (test) and cut-off points are also better described now
Line 141-149- Two-group and three-group comparisons were performed when selecting risk factors, but they are not described here. This information should be included.
We used Fisher’s and Chi-square tests to compared the prevalence of the groups. It is on the note of Figure 1, Table 1 and table 2. We also performed the Poison multivariate regression analysis to determine association. The multivariate model of regression analysis included variables with p-value ≤0.20 on simple regression but only the significates ones (p-value 0.05) are on the final model.
Results
The insertion position of figures and tables must be indicated in the text.
We modified it.
Line 240-245
Please provide the references.
We had included it.
Other
A similar study conducted in Brazil (reference 12) has found that cognitive function and malnutrition are risk factors. However, these were not examined in this study.
When we approved our study in the Ethical Committee the study (reference 12) was not publicised yet. We perform a literature review and we include all variables on our study based on scientific literature and also according our clinical expertise. However, we have in our study a variable of malnutrition which is low weight. In the table two you can observe several health variables that we had investigated.

Reviewer 2 Report
Comments and Suggestions for Authors
The manuscript titled "Sarcopenia in the oldest-old adults in the capital of Brazil: prevalence and its associated risk factors" presents a significant study on sarcopenia among older adults in Brasília, Brazil. The topic is highly relevant given the increasing aging population globally, particularly in Brazil. The study addresses a critical health issue—sarcopenia—which can significantly affect the quality of life in older adults.
There are minor comments:
Introduction
The study does not present new findings but reiterates known associations between sarcopenia and factors like gender and weight, which may limit its contribution to the existing literature.
The reliance on participants from specific health units may introduce selection bias, limiting the generalizability of the findings to the broader population of older adults in Brazil.
Methods
The methods section lacks comprehensive detail regarding participant recruitment and the exact procedures followed during assessments, which could affect reproducibility.
While multivariate Poisson regression is appropriate, the manuscript does not adequately justify the choice of variables included in the model, raising questions about potential confounders.
Discussion
The discussion heavily focuses on gender differences without adequately exploring other potentially significant factors, such as socioeconomic status or cultural influences.
The discussion section is somewhat superficial and does not sufficiently relate findings to existing literature or explore implications for public health policy.
The study fails to adequately address how comorbidities might interact with sarcopenia, which could provide a more nuanced understanding of risk factors.
Conclusion
The conclusion lacks clarity and does not effectively summarize key findings or offer actionable recommendations based on the results.
Author Response
We thank the reviewers for their thoughtful and in-depth comments concerning our manuscript. Your suggestions helped us improve the quality of our paper. We carefully considered every comment and made the appropriate changes, which are highlighted in red font.
Our point-by-point responses are also noted below
Introduction
The study does not present new findings but reiterates known associations between sarcopenia and factors like gender and weight, which may limit its contribution to the existing literature.
The reliance on participants from specific health units may introduce selection bias, limiting the generalizability of the findings to the broader population of older adults in Brazil.
We believe that including in our study individuals from health units can not introduce a relevant bias considering that a Brazilian study (reference 17) reveals that 46.2% of population over 60 years and older uses the health services. We know that individuals from 80 years and older in a country with high tax of violence do not accept who they don’t know on their homes. We believed that our recruitment strategy to include individuals who use the health system had reduced the bias of recuses. However, we do not found a study that have this information. We made this strategy based on our experience with this population with 80 years and older.
We included these information on Methods section, lines 88-93
Methods
The methods section lacks comprehensive detail regarding participant recruitment and the exact procedures followed during assessments, which could affect reproducibility.
We had improved these information in lines 123 and 142. We included more details of pre-sarcopenia, sarcopenia and severe sarcopenia and their respectives diagnostic criteria (test) and cut-off points.
Regarding the recruitment we also include more information:
“We know that individuals from 80 years and older in a country with high tax of violence do not accept who they don’t know on their homes. To minimize the recuses in our study and reduce bias we include individuals who uses the BHU. A Brazilian study reveals that 46.2% of population over 60 years and older uses the health services (17).”
While multivariate Poisson regression is appropriate, the manuscript does not adequately justify the choice of variables included in the model, raising questions about potential confounders.
We performed the Poison multivariate regression analysis to determine association. The multivariate model of regression analysis included variables with p-value ≤0.20 on simple regression but only the significates ones (p-value 0.05) are on the final model. More details are described on the last paragraph o Methods.
Discussion
The discussion heavily focuses on gender differences without adequately exploring other potentially significant factors, such as socioeconomic status or cultural influences.
The discussion section is somewhat superficial and does not sufficiently relate findings to existing literature or explore implications for public health policy.
The study fails to adequately address how comorbidities might interact with sarcopenia, which could provide a more nuanced understanding of risk factors.
Thank you for your recommendation. As our manuscript is an observational study the discussion should to focus on the variables which after multivariate regression to control potential confounders are statistically significant with the outcomes investigated, which were: for sarcopenia - female sex, low weight, ADLs; for severe sarcopenia - female sex, low weight and ADLs. Then we had discussed this association and we do not consider adequate to discuss variables which are not associated. It could be relevant in a systematic review study were could be explore serval types of variables that can contribute to sarcopenia. Other limitation to discuss more is the we almost do not found studies with this specific population – 80 years older and more to compare with our results.
We included a paragraph, lines 288-293/ 303-6
Conclusion
The conclusion lacks clarity and does not effectively summarize key findings or offer actionable recommendations based on the results.
According the recommendations of scientific writing rules, the conclusion should answer the objective of the study. Then it was that we wrote there. Our conclusion shows the occurrence of the studied outcomes and the associated factors.
We wrote recommendations of future studies and also clinical implication. It is between the lines 317-325.

Reviewer 3 Report
Comments and Suggestions for Authors
The study has potential given its large patient sample for sarcopenia assessment. It investigated the prevalence of sarcopenia and severe sarcopenia among Brazilians aged 80 and above, but the key risk factors associated with these conditions.
However, I have a few questions/concerns and potential suggestions for refinement.
Keywords
Please ensure keywords align with MeSH terms for optimized indexing.
Methods
The primary concern lies with the methodology, specifically the "sit-to-stand" test and the DXA measurements. Additionally, it’s unclear over what distance the gait speed was assessed; could you confirm adherence to guideline-recommended distances? 3or 4 meters?
To address missing data (i.e. DXA), you might reference Buccheri et al. (DOI: 10.1016/j.heliyon.2023.e16323), who developed a predictive model for screening muscle mass loss. This model uses only thigh and calf circumferences and has shown comparable accuracy to DEXA assessments for the diagnosis of muscle mass loss. You can add also another approach (DOI: 10.1519/JPT.0000000000000420) that suggests that body weight alone can approximate DEXA results with considerable accuracy, and accuracy further improves with the inclusion of sex and thigh/arm circumference in male patients. This additional context could help support your approach if data is limited.
Results
Consider adding a table summarizing the characteristics of patients with presarcopenia and sarcopenia, as this would clarify the definitions and help readers better understand the criteria and distinctions.
Author Response
Dear Reviewer.
We thank the reviewers for their thoughtful and in-depth comments concerning our manuscript. Your suggestions helped us improve the quality of our paper. We carefully considered every comment and made the appropriate changes, which are highlighted in red font.
Our point-by-point responses are also noted below
The study has potential given its large patient sample for sarcopenia assessment. It investigated the prevalence of sarcopenia and severe sarcopenia among Brazilians aged 80 and above, but the key risk factors associated with these conditions.
However, I have a few questions/concerns and potential suggestions for refinement.
Keywords - Please ensure keywords align with MeSH terms for optimized indexing.
Thank you. We had modified it.
Methods
The primary concern lies with the methodology, specifically the "sit-to-stand" test and the DXA measurements. Additionally, it’s unclear over what distance the gait speed was assessed; could you confirm adherence to guideline-recommended distances? 3or 4 meters?
Excuse but we didn’t use DXA or sit-to-stand test in this study. We had used 3 meters in the study and this information is between lines 139-143
To address missing data (i.e. DXA), you might reference Buccheri et al. (DOI: 10.1016/j.heliyon.2023.e16323), who developed a predictive model for screening muscle mass loss. This model uses only thigh and calf circumferences and has shown comparable accuracy to DEXA assessments for the diagnosis of muscle mass loss. You can add also another approach (DOI: 10.1519/JPT.0000000000000420) that suggests that body weight alone can approximate DEXA results with considerable accuracy, and accuracy further improves with the inclusion of sex and thigh/arm circumference in male patients. This additional context could help support your approach if data is limited.
Thank you for your concern. In our study muscle mass was assessed by calf circumference. This approach is consistent with previous works showing that anthropometric measurements are reliable to identify loss of muscle massin older adults. We inserted this reference in the study [25].
Results
Consider adding a table summarizing the characteristics of patients with presarcopenia and sarcopenia, as this would clarify the definitions and help readers better understand the criteria and distinctions.
Summarized data about presarcopenic and sarcopenic individuals are displayed at Table 1.
